# General Public’s Knowledge of Diabetes and Physical Activity in Saudi Arabia over Time: The Need to Refresh Awareness Campaigns

**DOI:** 10.3390/healthcare11030286

**Published:** 2023-01-17

**Authors:** Ghadah Alkhaldi, Naji Aljohani, Syed Danish Hussain, Hanan A. Alfawaz, Abdulaziz Hameidi, Gamal M. Saadawy, Mohamed A. Elsaid, Mohammed Alharbi, Shaun Sabico, Nasser M. Al-Daghri

**Affiliations:** 1Department of Community Health Sciences, College of Applied Medical Sciences, King Saud University, Riyadh 11451, Saudi Arabia; 2Obesity Endocrine and Metabolism Center, King Fahad Medical City, Riyadh 11525, Saudi Arabia; 3Biochemistry Department, College of Science, King Saud University, Riyadh 11451, Saudi Arabia; 4Department of Food Science and Nutrition, College of Food Science and Agriculture, King Saud University, Riyadh 11451, Saudi Arabia; 5Saudi Diabetes Charity, Riyadh 12721, Saudi Arabia; 6Diabetes Centres and Units Administration, Ministry of Health, Riyadh 11176, Saudi Arabia

**Keywords:** diabetes mellitus, physical activity, knowledge, awareness

## Abstract

Diabetes mellitus (DM) is a major health issue in Saudi Arabia. Prevention of DM and its complications requires an understanding of the disease and modifiable behaviors (e.g., physical activity—PA). The purpose of this study was to examine the trends in knowledge of the general population regarding DM to better understand the shortcomings in the current awareness programs. This article presents a cross-sectional series study where a survey was distributed to a total of 3493 participants over four years, from 2017 till 2020, to assess general knowledge about DM, including information about PA. The mean percentage of correct responses of DM general knowledge was 63.8 ± 19.0 in 2017, which decreased to 61.3 ± 18.7 in 2020 with a significant beta coefficient of −0.8 ± 0.2 (*p* < 0.001). Participants’ awareness about PA remained constantly high for four years: the mean percentage of correct responses was 82.1 ± 23.6 in 2017 and 82.0 ± 23.1 in 2020, and the beta coefficient was −0.5 ± 0.3 (*p* = 0.147). Furthermore, stratification by demographics showed that the majority of the subgroups (age, sex, educational status, marital status, having relative with DM, nationality) reported a significant declining trend in general DM knowledge. In addition, some of the subgroups also showed a declining trend in PA awareness. Future prevention efforts should assess the community’s DM knowledge regularly to tailor awareness efforts to the population segments that need heightened educational interventions.

## 1. Introduction

Diabetes mellitus (DM) is a major public health problem and a leading cause of death [1]. Half a billion people had DM worldwide in 2019, and half of those were not aware that they had it [2]. DM can be classified broadly into two types: type 1 (T1DM), which is an autoimmune disease characterized by complete deficiency of insulin secretion, and type 2 (T2DM), which accounts for 90% of DM cases worldwide, is due to insufficient insulin secretory response and resistance to insulin action [1].

According to the International Diabetes Federation (IDF), 17.7% of Saudi Arabia’s adult population suffers from DM, which is the second highest DM prevalence in the region and seventh worldwide [1]. Complications and mortalities related to DM affect individuals, communities and the whole country—Saudi Arabia has one of the highest rates of DM-related mortality and has a high percentage of health expenditure associated with DM morbidities [1,3]. Its prevention and reduction of its economic burden are two of the many strategical aims of the country [4]. Prevention of DM or its complications requires an understanding of the behaviors that could prevent such problems, including physical activity (PA), and a core understanding of the health problem itself. Increasing or enhancing knowledge is an essential component of any DM-related interventional effort. A systematic review of studies published in Saudi Arabia has found that poor knowledge about DM was a predictor of poor glycemic control [5]. Another recent study has shown that the Saudi general population suffers from a significant lack of knowledge about DM [6]. The importance of educating the general population rather than focusing on identified patients is due to the silent nature of the disease, whereby people are not aware of having it until they start to suffer from its complications [1]; the other reason is the importance of involving family members and friends in the care of people living with DM [7].

Numerous awareness-raising campaigns targeting the general population to prevent the development of DM and its complications have been implemented throughout the years in the Kingdom of Saudi Arabia [8]. Studies to measure people’s knowledge usually focus on a single point in time [5,6] or around the periods of major events [9], but given the lack of follow-up studies, it is unclear whether such programs were effective or not at increasing the people’s knowledge to in turn provide a clearer picture of whether awareness efforts need to change [10].

Riyadh is the capital city of Saudi Arabia, and more than 25% of the population lives in the capital alone [11,12]. It is the most urbanized city in the country and the one with numerous plans to increase its population size and invest in more lifestyle projects [13]. Demographically, Riyadh has certain factors that make it perfectly positioned to test and evaluate the effect of DM-related interventions; e.g., it is the city with the second highest number of elderly persons [14], and it has the largest number of citizens (and of residents) diagnosed with DM [15]. Hence, Riyadh makes a good case study for other cities within Saudi Arabia. One of the organizations leading these campaigns in Riyadh is the Saudi Diabetes Charity (SCAD), a local non-profit organization which has launched around 12,449 awareness campaigns since 2008 [8]. The charity conducts an annual evaluation of knowledge of the general population about DM and PA that started in 2017. The focus on knowledge about PA alongside DM due to the importance of PA in the prevention of DM and its management [16] and the government’s placing it in the forefront of interventional efforts [17], especially in Riyadh, for which there are numerous plans for building PA-conducive areas [13]. These plans, however, are faced with paucity in light of the still ongoing but plateauing COVID-19 pandemic.

The present study, we examine the trends in DM and PA knowledge amongst the general non-DM population in Riyadh, investigate any associations between demographic characteristics and DM and PA knowledge and identify demographics with a significant lack of knowledge that needs intensive educational interventions, granted the hypothesis that knowledge will them increase on both topics, especially for at-risk individuals.

## 2. Materials and Methods

### 2.1. Study Design

This was a cross-sectional series study using a self-report questionnaire that was distributed in paper format to the general adult population (≥26 years) not diagnosed with DM. The surveys were distributed annually from 2017 till 2020 through malls, community centers and other community outlets used by the Saudi Diabetes Charity when conducting awareness campaigns in Riyadh, Saudi Arabia.

Participants included in this study were adults above the age of 25 years old living in Saudi Arabia and not diagnosed with DM. Children, adolescents, young adults below 25 years old, DM individuals, mentally unwell people, those who could not read Arabic and those who did not consent were excluded from the study. Questionnaires were distributed at the beginning of awareness activities held across Riyadh. All the participants were informed about the purpose of the study and signed a consent form before filling out the questionnaire. The study was approved by the Institutional Review Board (IRB) of the College of Medicine, KSU, Saudi Arabia (No. E-19-4239, 29).

### 2.2. Sample Size

With an estimated mean of correct response of 78.15% among non-diabetic subjects in the previous study [18], this study required a sample size of 1954 participants (95% CI with 2% margin of error). However, this study recruited a total of 3493 participants over 4 years.

### 2.3. Instrument Description

The questionnaire was sent to experts for content and face validity. Furthermore, the internal consistency of the questionnaire was evaluated using Cronbach’s alpha, which was above the required threshold of 0.70. The Cronbach’s alpha (α) values were 0.77 and 0.71 for DM general knowledge and PA, respectively.

The questionnaire’s items included in the analysis were: demographic information, general knowledge about diabetes—causes, symptoms and characteristics, awareness of risk factors, health risks and prevention associated with DM and knowledge about PA. (See Appendix A).

### 2.4. Data Analysis

Data were analyzed using IBM SPSS Statistics for Windows, Version 22.0 (IBM Corp., Armonk, NY, USA). Normality of the variables was evaluated based on absolute skewness value ≤ 2 and absolute kurtosis (excess) ≤ 4. Continuous variables are presented as mean ± standard deviation, and categorical variables are presented as frequencies and percentages. Multivariate analysis of variance (MANOVA) was used to determine the significant differences between various categories regarding general knowledge about DM and PA. Linear regression analysis was used to assess the trends in DM and PA knowledge scores over the years with year as the independent variable. Homogeneity and homoscedasticity were evaluated using a scatterplot of the residuals against the predicted values of the dependent variable. The results were in line with the homogeneity assumption of linear regression analysis. Furthermore, a Mantel–Haenszel test of trend was run to determine whether a linear association existed between correct or incorrect responses. *p* < 0.05 was considered as significant.

## 3. Results

### 3.1. Sample Description

Over the four years, a total of 3493 participants filled out the surveys. Most of those were between 26–35 years old (46.2%), males were the majority (60.6%) and 79.2% were married. The majority of participants were Saudis (73.7%) and had a bachelor’s degree (50.9%). Around 70% of them had a relative diagnosed with DM (Table 1).

### 3.2. Differences in DM and PA Knowledge According to Demographics

Mean correct responses of DM general knowledge and PA according to demographics are presented in Table 2. The results show that older participants had higher correct responses for both DM general knowledge (*p* < 0.001) and PA awareness (*p* < 0.001). Female participants had significantly more correct responses on DM general knowledge (*p* < 0.001) but had fewer correct responses on PA awareness (*p* = 0.003) than their male counterparts. Married participants had more correct responses that single ones for both DM general knowledge (*p* < 0.001) and PA awareness (*p* = 0.003). Divorced participants also had significantly more correct responses on DM general knowledge as compared to singles (*p* < 0.001). Diploma holders had significantly fewer correct responses on both DM general knowledge (*p* < 0.001) and PA awareness (*p* = 0.003) than either bachelor’s or master’s degree holders. Participants who had relatives with DM had significantly more correct responses for both DM general knowledge (*p* < 0.001) and PA (*p* < 0.001) awareness than participants with no relatives with DM. Furthermore, Saudi nationals also had significantly more correct responses on PA awareness (*p* < 0.001) than non-Saudis.

### 3.3. DM and PA General Knowledge by Topic over the Years

There was a consistent decline in the percentage of correct responses over the four years for the following items related to DM knowledge: insulin shortage as a cause of DM, T1DM is a chronic uncurable condition and screening for DM (See Appendix A).

### 3.4. DM and PA General Knowledge over the Years

Table 3 shows the mean correct responses and beta (β) coefficients for DM general knowledge and PA over the years. Beta coefficients are obtained from linear regression analysis with year as the independent variable.

The mean percentage of correct responses for DM general knowledge was 63.8 ± 19.0 in 2017, which decreased to 61.3 ± 18.7 in 2020 with a significant beta coefficient of −0.8 ± 0.2 (*p* < 0.001). There was a similar significant declining trend in DM general knowledge in various demographics, with the exceptions being females (*p* = 0.65), married people (*p* = 0.30), divorced people (*p* = 0.43), those with bachelor’s degrees (*p* = 0.19), those with no diabetic relatives (*p* = 0.32) and those who were non-Saudi’s (*p* = 0.84). Participants older than 45 showed a decline in DM general knowledge but with borderline significance (*p* = 05). The only subgroup which had a significant increase in DM general knowledge was the widowed participants (*p* = 0.01).

Participants’ awareness about PA remained constantly high for four years: the mean percentage of correct responses was 82.1 ± 23.6 in 2017 and 82.0 ± 23.1 in 2020, and the beta coefficient was −0.5 ± 0.3 (*p* = 0.15). Furthermore, stratification by demographic variables showed that majority of subgroups reported no change in PA awareness, exceptions being male participants (*p* < 0.001), single (*p* = 0.017), diploma holders (*p* = 0.009) and master’s degree holders (0.039).

Individual item analysis of DM general knowledge indicated that participants were least likely to respond correctly to the following items in 2017: “DM among women is higher than among men.” “In healthy individuals, blood sugar levels are considered high if the blood sugar level over 100 mg/dcl (fasting).” “A random blood sugar level over 200 mg/dcl is an indication of having DM.” “Fundoscopy and a retinal exam for DM patient are recommended only when necessary.” “DM is considered as main cause of heart disease.” The rates of correct responses were 32.0%, 42.5%, 43.1%, 45.6% and 48.9% in 2017 respectively. Participants’ responses to these items in 2020 were correct 25.9%, 35.9%, 40.4%, 51.1% and 47.5% of the time, respectively. Furthermore, a significant decline in correct responses over the years was observed for the following items: “Obesity causes DM.” “Full or partial shortage of insulin in the blood causes DM.” “Type 1 DM is a chronic disease with no available cure for the time being, but it can be managed.” “In healthy individuals, fasting blood sugar levels are considered high if above 100 mg/dcl.” “Eye function could be affected by chronic high blood sugar levels and may lead to blindness.” “Uncontrolled type 1 DM patients should not practice fasting.” “Polyuria (frequent urination) during night, sudden/unexplained weight loss and excessive thirst are symptoms of hyperglycemia.” “Gestational diabetes mellitus may increase the risk of giving birth to an overweight baby and might lead to her/his death.” “Screening for DM is at the age of 40, or earlier if risk factors exist.” “More women than men are diagnosed with DM.” “Foot amputation is mainly caused by DM.” “Fundoscopy and retinal exam for DM patients are recommended only when necessary.”

Individual item analysis of PA awareness indicated that participants were correct more than 50% of the time on all items in 2017. Furthermore, a significant increase in correct responses over the years was observed for the following items: “Walking as exercise has health benefits for DM patients.” “Increasing daily PA is important for DM management” (See Appendix A).

## 4. Discussion

This is the first study showing the trends of DM and PA knowledge among citizens and residents living in Riyadh, Saudi Arabia, and it had a larger sample size than previously published studies [19]. Knowledge about diabetes has stayed relatively high over the four years of the study, as the mean percentage of correct responses was above 50% for all categories. However, there was a consistent declining trend, especially for those under 45 years old, males, singles, and/or those with a relative with DM. Specific knowledge about PA stayed high; there was no significant change over the years. A positive, upward trend for knowledge about DM and PA was seen for people with no official education. However, the differences between groups showed that those over 45 years old, married people, those with a relative with diabetes, and Saudis had significantly more knowledge, compared to their respective counterparts, about PA and DM. Additionally, the mean percentage of correct responses for males for PA was significantly higher than for females but lower for DM.

Compared to a recent study that looked at the general population’s knowledge of DM in Saudi Arabia, this study has shown that education and age are associated with a change in people’s knowledge about DM and that the knowledge level, although declining, is above 50% [6]. The difference in the results could be due to the way knowledge was measured in the studies, using different questions, judging knowledge based on awareness of specific DM-related topics and the exclusion of those with known DM diagnosis in this study. In addition, this study was conducted only in the city of Riyadh, the capital of Saudi Arabia, and a city with one of the highest rates of DM diagnosis [15]. A recent study performed in Singapore [20] showed results consistent with the present study: older age was associated with better knowledge of diabetes, and being male was associated with a lower level of knowledge. This is not surprising, as numerous studies have shown that certain demographic and personal factors [21,22,23] are predictors of DM (such as education, age, gender and having a relative with DM) and that tailored educational interventions are needed for such groups to delay or prevent the development of DM [24,25,26,27].

DM knowledge studies performed in Saudi Arabia have shown that the population has a low level of knowledge about risk factors for DM and DM complications [6,19], and this remained consistent in some of our results, whereby specific DM information that was increasingly answered incorrectly included the screening age for DM.

DM awareness and knowledge among those at risk or the general population can decline due to several possible reasons: people tend to not actively seek DM-related information [28], not being able to apply information given due to low health literacy [29], the focus of awareness campaigns on motivation as the single factor for influencing the different behaviors related to DM (e.g., screening, diet and PA) instead of addressing other factors that enable people to change their behaviors, such as capabilities and opportunities [10], and not tailoring the message or channel of the campaign to a targeted audience but generalizing it will not resonate with the people [30].

PA knowledge did not change over time, but the variance in knowledge about PA was higher for males compared to females. This might have been due to the fact that a lot of PA initiatives in Saudi Arabia focused on males, though one review recommended providing more opportunities for females [31].

With the Kingdom’s 2030 vision [4] focusing on prevention of chronic disease, and diabetes being one of the leading chronic diseases in SA [1], national educational and awareness programs and interventions need to address the gaps in people’s knowledge using channels and messages tailored to the population segments that need more targeted education efforts. These educational efforts need to be joined with policy actions to be more effective [30] and to focus on messages that provide practical information and take into consideration not only motivating the people to change, but to maintain behavioral change [10]. Studies assessing knowledge of a community should use a unified and validated knowledge questionnaire to facilitate the production of concrete and evidence-based recommendations for public health interventions [32].

The study had limitations. The sample was a convenience sample, which might have resulted in the exclusion of certain groups that did not attend community hubs, as opposed to doing a household survey. The cross-sectional nature of the survey does not allow for casual inference, but the repeated nature of the study enabled the observation of any changes overtime. The study was limited to the city of Riyadh; however, Riyadh is perfectly positioned to be a case study for the rest of Saudi Arabia, since it is the focus of major urbanized development and lifestyle interventions [13] and has the largest proportion of the population of any city in the country [12], especially of those at risk [14]. Lastly, information from the study was obtained using self-reported responses from questionnaires.

Despite the limitations, the findings of the study have substantial clinical relevance in terms of improving the general public’s health through raising DM awareness. Recent studies have already raised the alarm on the increasing prevalence of pediatric metabolic syndrome over time in Saudi Arabia [33]. The impact of COVID-19 lockdowns has also exacerbated the burden of an already prevalent sedentary lifestyle within the Saudi population [34,35,36,37]. These findings, together with the present study, collectively point to unfavorable public health status that is incompatible with and even detrimental to the Saudi Vision 2030 if left on its own to further deteriorate. The present paper recommends, therefore, that health authorities and health policy makers take these observations seriously and start revitalizing the once active DM awareness campaigns through different public and private sectors. Increased mobilization and outdoor activities during pre-pandemic levels should again be encouraged, targeting populations at risk, such those who are obese, the elderly, non-Saudis and families with high susceptibility to DM.

## 5. Conclusions

There is a general declining trend in the level of DM knowledge and no change in that of PA. The study identified certain at-risk groups in Saudi Arabian citizens and residents that require intensive DM educational and awareness campaigns with a focus on the knowledge that could personally impact them. Future prevention interventions should assess the community’s knowledge regularly to tailor efforts to the population segments in need.

## Figures and Tables

**Table 1 healthcare-11-00286-t001:** Sample demographics (*n* = 3492).

Demographic Characteristics	*n* (%)
Age (Years)	
26–35 36–45 46–55 56–65 Over 65	1615 (46.2) 1266 (36.2) 471 (13.5) 124 (3.5) 17 (0.5)
Sex	
Male Female	2088 (60.6) 1360 (39.4)
Marital Status	
Single Married Divorced Windowed	540 (15.7) 2719 (79.2) 141 (4.1) 33 (1.0)
Nationality	
Saudi Non-Saudi	2255 (73.7) 804 (26.3)
Education	
Diploma Bachelors Masters	1229 (35.2) 1777 (50.9) 487 (13.9)
Do you have any relative with DM?	
Yes No	2216 (70.0) 949 (30.0)

Note: Data presented as *n* (%).

**Table 2 healthcare-11-00286-t002:** DM and physical activity knowledge according to demographics for the four years combined.

Demographic Characteristics	Physical Activity		DM General Knowledge	
Age	Mean ± SD	*p*-Value (Effect Size)	Mean ± SD	*p*-Value (Effect Size)
<45 Years	81.0 ± 23.8	<0.001 (0.005)	62.2 ± 18.8	<0.001 (0.009)
≥45 Years	86.2 ± 21.5		68.3 ± 18.1	
Sex				
Male	82.7 ± 24.6	0.003 (0.002)	62.4 ± 20.1	<0.001 (0.004)
Female	80.5 ± 22.4		63.9 ± 17.1	
Marital Status				
Single	78.8 ± 24.5 ^A^	0.003 (0.004)	58.8 ± 18.6 ^AB^	<0.001 (0.026)
Married	83.7 ± 22.6		65.8 ± 18.2 ^B^	
Divorced	79.9 ± 25.2		64.1 ± 20.0	
Windowed	80.0 ± 21.9		62.5 ± 18.8	
Education				
Diploma	80.3 ± 25.4 ^AB^	0.003 (0.004)	61.4 ± 19.8 ^AB^	<0.001 (0.009)
Bachelors	83.4 ± 20.8		65.0 ± 16.5	
Masters	84.3 ± 21.9		65.0 ± 18.0	
Relatives with DM				
Yes	83.1 ± 22.0	0.001 (0.003)	64.4 ± 17.8	<0.001 (0.008)
No	79.5 ± 25.7		60.2 ± 19.8	
Nationality				
Saudi	82.9 ± 22.4	<0.001 (0.006)	63.5 ± 18.3	0.261 (0.000)
Non-Saudi	79.2 ± 25.9		61.9 ± 20.1	

Note: Data are presented as mean percentages of correct responses; superscript ^A^ and ^B^ indicate significance from 2nd and 3rd categories respectively; *p*-values were obtained from MANOVA; *p* < 0.05 is considered significant.

**Table 3 healthcare-11-00286-t003:** Trends in DM and physical activity knowledge according to demographics and year.

Demographic Characteristics	2017	2018	2019	2020	Beta ± SE	*p*-Value (R^2^)
All Participants						
Physical Activity	82.1 ± 23.6	82.5 ± 23.0	79.4 ± 25.3	82.0 ± 23.1	−0.5 ± 0.3	0.15 (0.00)
DM General Knowledge	63.8 ± 19.0	63.7 ± 18.5	62.2 ± 19.1	61.3 ± 18.7	−0.8 ± 0.2	0.001 (0.002)
Age: <45 Years						
Physical Activity	81.4 ± 23.8	81.9 ± 23.0	78.2 ± 25.7	81.6 ± 22.7	−0.4 ± 0.3	0.21 (0.000)
DM General Knowledge	63.0 ± 19.0	62.8 ± 18.4	61.0 ± 19.1	60.7 ± 18.3	−0.8 ± 0.3	0.002 (0.002)
Age: >45 Years						
Physical Activity	87.3 ± 18.4	86.6 ± 22.2	85.0 ± 22.2	84.8 ± 24.5	−1.0 ± 0.8	0.21 (0.002)
DM General Knowledge	69.4 ± 16.2	69.0 ± 18.9	68.3 ± 17.8	65.1 ± 20.2	−1.3 ± 0.6	0.05 (0.006)
Sex: Male						
Physical Activity	84.1 ± 23.3	83.7 ± 23.6	77.7 ± 28.9	83.3 ± 23.7	−1.1 ± 0.4	0.01 (0.002)
DM General Knowledge	64.1 ± 19.5	62.4 ± 19.8	59.8 ± 21.4	60.8 ± 20.1	−1.4 ± 0.4	<0.001(0.006)
Sex: Female						
Physical Activity	79.0 ± 23.3	81.5 ± 22.3	81.1 ± 21.3	80.6 ± 22.4	0.6 ± 0.4	0.20 (0.001)
DM General Knowledge	63.2 ± 18.0	64.9 ± 17.1	64.6 ± 16.2	62.1 ± 16.5	−0.2 ± 0.3	0.65 (0.000)
Marital Status: Single						
Physical Activity	80.7 ± 23.7	80.3 ± 23.3	72.4 ± 27.5	80.6 ± 23.3	−1.3 ± 0.6	0.02 (0.003)
DM General Knowledge	61.2 ± 19.3	59.2 ± 17.8	55.8 ± 18.3	56.6 ± 18.6	−1.9 ± 0.4	<0.001 (0.011)
Marital Status: Married						
Physical Activity	83.2 ± 23.0	84.3 ± 22.4	84.1 ± 22.6	83.3 ± 22.3	0.1 ± 0.4	0.77 (0.000)
DM General Knowledge	65.6 ± 18.2	66.7 ± 18.0	66.1 ± 18.6	64.1 ± 17.9	−0.3 ± 0.3	0.30 (0.000)
Marital Status: Divorced						
Physical Activity	81.5 ± 25.2	78.8 ± 26.3	77.9 ± 25.7	83.3 ± 22.5	0.0 ± 1.9	0.98 (0.000)
DM General Knowledge	67.3 ± 20.5	62.2 ± 21.3	63.9 ± 17.2	62.9 ± 20.5	−1.2 ± 1.5	0.43 (0.004)
Marital Status: Widow						
Physical Activity	77.6 ± 14.5	86.0 ± 16.9	79.5 ± 34.8	77.8 ± 32.3	0.4 ± 2.4	0.86 (0.000)
DM General Knowledge	53.2 ± 15.7	73.9 ± 11.5	68.9 ± 23.3	64.0 ± 20.1	5.1 ± 2.0	0.01 (0.083)
Educational Status: Diploma						
Physical Activity	80.9 ± 25.6	84.0 ± 22.2	75.4 ± 28.1	79.0 ± 25.3	−1.5 ± 0.6	0.009 (0.004)
DM General Knowledge	62.2 ± 20.4	64.0 ± 18.7	59.2 ± 19.9	58.4 ± 19.7	−1.5 ± 0.5	0.001 (0.006)
Educational Status: Bachelor’s Degree						
Physical Activity	83.1 ± 20.7	82.9 ± 21.3	84.0 ± 19.6	84.4 ± 21.0	0.5 ± 0.4	0.24 (0.001)
DM General Knowledge	66.1 ± 16.1	63.7 ± 17.2	66.0 ± 15.9	64.1 ± 16.6	−0.4 ± 0.3	0.19 (0.001)
Educational Status: Master’s Degree						
Physical Activity	86.7 ± 19.1	83.7 ± 22.2	81.1 ± 26.6	82.4 ± 23.1	−1.8 ± 0.8	0.04 (0.000)
DM General Knowledge	66.1 ± 16.6	68.1 ± 17.3	62.1 ± 20.2	59.9 ± 19.3	−2.1 ± 0.7	0.003 (0.017)
DM Relative: Yes						
Physical Activity	83.6 ± 22.1	83.8 ± 21.2	79.9 ± 24.0	83.9 ± 20.4	−0.5 ± 0.4	0.20 (0.001)
DM General Knowledge	65.1 ± 18.5	64.9 ± 17.7	63.7 ± 17.1	62.4 ± 16.7	−0.8 ± 0.3	0.007 (0.002)
DM Relative: No						
Physical Activity	78.7 ± 25.2	78.8 ± 27.1	80.0 ± 26.1	82.0 ± 23.9	1.0 ± 0.6	0.12 (0.002)
DM General Knowledge	59.6 ± 18.8	60.2 ± 20.3	60.6 ± 21.0	61.1 ± 19.9	0.5 ± 0.5	0.32 (0.001)
Nationality: Saudi						
Physical Activity	83.9 ± 21.6	82.7 ± 22.7	81.7 ± 22.8	82.7 ± 22.7	−0.5 ± 0.4	0.13 (0.001)
DM General Knowledge	64.7 ± 18.1	63.5 ± 18.7	63.7 ± 17.9	61.6 ± 18.4	−0.9 ± 0.3	0.003 (0.003)
Nationality: Non-Saudi						
Physical Activity	77.4 ± 27.5	80.7 ± 23.8	80.7 ± 26.8	78.7 ± 23.0	0.9 ± 0.8	0.26 (0.001)
DM General Knowledge	61.8 ± 21.1	62.1 ± 18.6	63.0 ± 20.6	60.0 ± 19.2	−0.1 ± 0.6	0.84 (0.000)

Note: Data presented mean percentages of correct responses; beta and standard errors are obtained from linear regression; (R^2^) indicates effect size; *p* < 0.05 is considered as significant.

## Data Availability

The data presented in this study are available on request from the corresponding author. The data are not publicly available due to privacy protection.

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
