# Peer review of "General Public’s Knowledge of Diabetes and Physical Activity in Saudi Arabia over Time: The Need to Refresh Awareness Campaigns"

_healthcare, 2023, doi:10.3390/healthcare11030286_

Round 1

Reviewer 1 Report (Previous Reviewer 1)

Thank you for addressing the suggestions given.

Author Response

Thank you for addressing the suggestions given.

Response: We thank the reviewer for appreciating our revisions.

Reviewer 2 Report (Previous Reviewer 2)

Thank to the authors for the efforts spent in addressing the raised issues and improving the manuscript according to suggestions. Despite the low novelty of the proposed approach, the study is scientifically sound and has a significant social value. The changes in the introduction and discussion made the contextualization and the validity of the study more clear. However, there is still need for a proofreading of the manuscript in order to correct language and editing mistakes; please, pay particular attention to the use of punctuation. 

Author Response

Thank to the authors for the efforts spent in addressing the raised issues and improving the manuscript according to suggestions. Despite the low novelty of the proposed approach, the study is scientifically sound and has a significant social value. The changes in the introduction and discussion made the contextualization and the validity of the study more clear. However, there is still need for a proofreading of the manuscript in order to correct language and editing mistakes; please, pay particular attention to the use of punctuation. 

Response: We thank the reviewer for appreciating our first revision. We have taken into full consideration the suggested editorial changes for added clarity.

Reviewer 3 Report (Previous Reviewer 3)

Thank you for the revision. As the footnote of Table 2 shows, it seems that you conducted a multivariate regression analysis. In this case, MANOVA was not conducted, and you should rewrite MANOVA to multivariate regression model in the Methods.

Author Response

Thank you for the revision. As the footnote of Table 2 shows, it seems that you conducted a multivariate regression analysis. In this case, MANOVA was not conducted, and you should rewrite MANOVA to multivariate regression model in the Methods.

Response: We thank the reviewer for this comment. We confirm that MANOVA was conducted in table 2 and revised the footnote accordingly. Linear regression analysis was done in table 3.

This manuscript is a resubmission of an earlier submission. The following is a list of the peer review reports and author responses from that submission.

Round 1

Reviewer 1 Report

Nice study.  I like the idea of refreshing societies mind about DM and PA’s effects.  Good job. 

One suggestion you may want to think about is giving ways to get the word out about DM and PA to society.  I think this would be beneficial to health educators throughout the world.  

Reviewer 2 Report

The proposed paper “Recent Trends in the General Public’s Knowledge of Diabetes and Physical Activity in Saudi Arabia: The Need to Refresh Awareness Campaigns” analyzes the answer to a questionnaire submitted to 3493 people in the city of Riyadh, related to subject’s knowledge about Diabetes Mellitus and the impact of physical activity on health.

Despite the social value of the topic, the proposed study does not present any innovation, either in the methodology or in the results, and the research gap filled is limited to the very specific situation of the city of Riyadh, with no clues about possible extendibility.

Moreover, there are some technical issues that require attention, the most important being:

1) Line 117 <<Continuous variables were presented as mean ± standard deviations>>: the use of mean and standard deviation, as well as the ANOVA test, are only applicable to normal distributions. It is necessary to assess the normality of the distribution with a dedicated test, and, in case the distribution results non-normal, data should be presented as median, 25th – 75th percentile, and a non-parametric test should be used instead of ANOVA (as for example Wilcoxon’s rank-sum test with Bonferroni correction for multiple groups).

2) Line 133: It seems table 2 is referred in the text as table 3, meaning that table 3 is not referred in the text. Please check tables numbering, and make sure they are positioned after their first reference in the text.

Other minor issues are:

3) Line 53: comma between subject and predicate.

4) Lines 90-114: For a better readability, I suggest presenting the questionnaire sections, and related questions, in a table rather than as plain text.

5) Line 160: ‘yeas’ instead of ‘years’

6) Line 202-205: part of the template is left in the manuscript.

7) Line 220: ‘but’ may be a typo.

Reviewer 3 Report

The study seems to be conducted well. I have some minor comments.

1.         (Abstract) 3493 respondents in total were survey in 4 years or 3493 respondents were surveyed every year during the four years?

2.         (Abstract, line 26) The authors need to mention about regression analysis as a method before writing beta coefficients as a result. In addition, please write about statistical analyses used in this study.

3.         (Abstract, line 29-31) Specifically, what kind of subgroups were used?

4.         (Methods, lines 95) What “must not fast” mean?

5.         (Methods) Could you attach the questionnaire used in this study as a supplementary material?

6.         (Methods, lines 122-123) It is not certain how did you use linear regression and Mantel-Haenszel test in this study. I think that both analyses are used for analyzing the trend of proportion of correct, but why did you use both of them?

7.         (Table 2) It might be better to conduct a multivariate analysis rather than bivariate analysis by t-test. You can do logistic regression or other analysis easily by SPSS.

8.         (Discussion) Could you write a possible reason why declining trend in the levels of DM knowledge was observed in Discussion?